# Crip Linguistics Goes to School

**Jon Henner [1],\* and Octavian Robinson [2]**

1   Interpreting, Deaf Education, and Advocacy, Specialized Education Services, University of North Carolina Greensboro, Greensboro, NC 27403, USA

2   Deaf Studies Department, Gallaudet University, Washington, DC 20002, USA

\*   Correspondence: j_henner@uncg.edu

**Abstract:** Teachers of the deaf, signed language interpreters, and associated staff (e.g., secretaries and sanitation workers) are a salient part of a deaf child's community often because hearing parents and other family members of deaf children do not become proficient signers leading many deaf children and adults to feel alienated in the home environment (e.g., dinner table syndrome). Because the school environment provides another way for deaf children to acquire language, professional signed language fluency is critical. Yet, in other second language acquisition contexts, fluency is not necessary for effective teaching and often highly racialized. If perceived fluency is often dependent on proximity to whiteness, and language fluency is not necessary for effective teaching, then why is it necessary to require professionals to be fluent in signed languages before teaching and working with deaf children?

**Keywords:** Crip Linguistics; translanguaging; raciolinguistics; language deprivation; signed languages

## 1. Introduction

*Jon*: A few weeks before I started this document, I hosted friends of mine that I have known for several decades. We have children of similar age. We turned on a new-to-us show wherein songs were translated into American Sign Language (ASL) by a series of performers to keep our children occupied. After watching the show together for several songs, I noted to myself that none of the performers were fluent in ASL. However, we left the show on for the children because accessible media on streaming networks is extraordinarily rare. Later that evening, I reflected on the show, my thoughts about the signers, and how the idea of fluent signing fits into the principles of Crip Linguistics.

*Octavian*: When Jon approached me with this paper, I wondered about the same. As a faculty member at Gallaudet University, where the majority of students are deaf, deafblind, or deafdisabled, a part of our mission is to develop communicative competencies. The process of developing communicative competencies raised questions about expectations surrounding proficient signing for both students and faculty. I reflected on questions of fairness in assessments and expectations for signing proficiency for academic performance for students and for tenure and promotion decisions for faculty within the principles of Crip Linguistics. If Crip Linguistics asserts that no way of languaging is wrong or broken, then what is our aim in expecting and assessing fluency?

The three major principles of Crip Linguistics are:

1. Language is not inherently disordered, although impairments may exist;
2. Deficit perceptions of the body–mind disorders language uses;
3. Disability in languaging cannot be separated from normative expectations of language use.

Crip Linguistics embraces linguistic variation while rejecting normative expectations of language use by adopting an activist stance rooted in critical disability politics. To crip

language is to embrace disabled ways of languaging. For further discussion on *crip* as a verb in the context of languaging and our usage of languaging as a verb, (see Henner and Robinson, in press). Linguistic fluency is wrapped up in concepts of ability and whiteness. Fluency appears to be at odds with the principles of Crip Linguistics. So, is it *crip* to ignore fluency in critical language education areas such as teaching? Can one be *crip* and require fluency in others?

The goal of this paper is to discuss the contradiction of required signed language fluency in teachers of the deaf under the Crip Linguistics framework. To achieve this, we discuss the challenges to the concept of *fluency* because the term is laden with racist, classist, and ableist notions of what makes good language. As both authors are situated in the United States, we use our local standardized signed language, American Sign Language (ASL), in most of our examples, especially with relationship to the standard English used widely. However, many of our examples are applicable to the dynamics between signed and spoken languages globally. We also question if it is possible to assess for fluency in both teachers and students without creating an environment wherein language is being judged by racist, ableist, and sexist frameworks. Yet, requiring fluent signers for deaf students is a disability justice issue. This paper explores what makes for fluency and proficiency in the context of Crip Linguistics, which acknowledges the stark realities of language deprivation and linguistic capital, where linguistic capital is defined as access to a community of language users (e.g., Listman et al. 2011).

Deaf children largely experience language deprivation and absence of linguistic capital. Many deaf children do not have access to a community of languagers outside of school. Approximately 90–95 percent of deaf children are born into non-signing families. Deafness itself is a very low-incidence disability with few exceptions in some areas with unusual genetic incidence of deafness, such as Bengkala, Bali, Adamorobe, Ghana, or Martha's Vineyard of the 19th century. Because of the overall low incidence of deafness, there is also not much material in signed language available in the form of children's literature and television. Therefore, most deaf children grow up around non-signing hearing people within their families, local communities, and digital spaces. Local community members are typically not signers, and signing schools for the deaf are usually not the first destination for deaf children. In van der Straaten et al.'s (2021) work it has been shown that the vast majority of deaf children in the Global North attend "inclusive" schools where, if they sign, they are assigned signed language interpreters. Signed language interpreters are often the only or "most fluent" sign language model the child is exposed to, but they have variable fluency and are not a substitute for direct instruction or socialization (De Meulder and Haualand 2021; Caselli et al. 2020). There is a clear need for fluent and proficient signing models for deaf children to ensure full social, political, and cultural participation, as well as achieve emotional, mental, and physical wellbeing (Murray 2015; Hall 2017).

To make this argument, we reintroduce *Crip Linguistics* as it is the theoretical foundation of this paper. The three main relevant theories that undergird Crip Linguistics are language deprivation, translanguaging, and raciolinguistics. Language deprivation discourses demonstrate why deaf children need to be in good language environments; language deprivation requires that we understand that deaf children often do not have the opportunity to learn language like hearing children do. Translanguaging reflects the reality of how people use languages in opposition to popular ideas about how languages function. We include raciolinguistics because our idea of good language is heavily structured on white supremacy in the Global North. Finally, we wrap this paper up by returning to the two main ideas that we are arguing and demonstrating how each point can be performed. First, to understand why this argument needs to be made, we need to understand language attitudes in the context of signed languages, present-day conditions for deaf children, the poor state of deaf education, and the linguistic complications that accompany deaf education.

Language deprivation is driven by negative attitudes toward signed languages. It is a known quality in deaf communities that hearing people with varying proficiency in

signed languages often appropriate signed languages for attention, for jobs, and because they believe signed languages are "lesser-than" and not deserving of the same respect that spoken languages receive (Robinson and Henner 2017). Negative language attitudes have led to the decimation of direct education in signed languages. Murray et al. (2020) estimate only 2% of all deaf children worldwide are exposed to signed language during formative developmental periods. In truth, the number of signing deaf children decreases because of institutional pushes for the removal of any kind of signed language use and education of signed language among deaf children lest spoken language skills be impacted (see Geers et al. 2017; and more recently Tamati et al. 2022). The fact that access to signed languages is a human right (e.g., Snoddon and Murray 2019) is simply a non-issue for people and policymakers who would eradicate signed languages from deaf education. Nevertheless, talking about deaf signers is critical because studying how they learn and use language has the potential to add to our knowledge of language development and socialization. A critical factor in language development of deaf signers, because of the lack of community language use, is the presence (or absence) of fluent signers in the deaf education classroom. Yet, how fluency is perceived is moderated by racism. As there is limited discussion on how racism impacts signed language perception and assessment, we must use spoken language theories.

Crip Linguistics is intrinsically entangled with raciolinguistics. Raciolinguistics is a field of study in which researchers examine how racialization and language interact. Language, as an embodied action, cannot be separated from the bodies that are read as deficient or non-normative in some way. Because non-white bodies are viewed as deficient, they experience ableism even if they do not have an impairment. The logics of racism and ableism, given the historical relationships between race and disability, are closely intertwined and cannot be separated (Fish 2019). That is, we cannot address racism without also addressing ableism and vice versa. As Talila Lewis reminds us, ableism is fundamentally about identifying and repairing non-normative bodies (https://twitter.com/talilalewis/status/1563588651400826882, accessed on 25 January 2023). If whiteness is considered the norm, then non-white people are labeled deficient based on the perception of their distance from whiteness, which in itself is a dynamic historical category (Omi and Winant 2020) On this, Flores and Rosa (2015) write:

> *Specifically, we argue that the ideological construction and value of standardized language practices are anchored in what we call raciolinguistic ideologies that conflate certain racialized bodies with linguistic deficiency unrelated to any objective linguistic practices. That is, raciolinguistic ideologies produce racialized speaking subjects who are constructed as linguistically deviant even when engaging in linguistic practices positioned as normative or innovative when produced by privileged white subjects. (p. 150)*

This means that people seen as racialized, which here in the United States are usually people who are not perceived as white, will almost always have language considered deficient by people assessing language. To simplify, black and brown speakers are often not seen as *fluent* users of their own languages. However, people close to whiteness using the same language are seen as innovative. For example, Ilbury (2020) describes how white gay men are rewarded for creating language personas based on sassy black women in ways that black women would not. Although gay men are themselves marginalized for their sexuality, their whiteness gives them the ability to cosplay as sassy black women without the same kind of social stigma that is applied to black women. Signed languages are subject to the same racialized politics that spoken languages are subjected to; intellectual histories of signed languages demonstrate that negative attitudes about signed languages are rooted in perceptions of signed languages as an expression of racial inferiority. Additionally, deaf people occupy all kinds of bodies, including black and brown bodies, and their signing is also subjected to racist logics. Those logics play a role in language deprivation.

Discussions about the absence of teacher signed fluency in deaf education are not new, although there is little to no discussion on how racism plays into this because deaf education researchers historically saw deaf children as an amorphous white blob. Both Stewart (1993)

and Akamatsu et al. (2002) argued that signed language skills were not necessary for educating deaf children. For them, it was more important that teachers be good teachers; signing skills came second. Through the years, the argument against the need for teachers to sign well did not change much. Stewart (1993) thought that ASL was much too hard for teachers to learn and that requiring teachers to be proficient in natural signed languages, as opposed to constructed signed systems such as Signed Exact English, would stop them from using what would later be called their semiotic repertoire via calibration (Kusters et al. 2017; Moriarty and Kusters 2021). The early concept of the semiotic repertoire and calibration was included in Akamatsu et al. (2002), where they discussed how good communication required working with their students to understand them and to be understood. As the students in question were often novice signers, Akamatsu et al. felt that teachers enforcing and using ASL would be a violation of communicating with the students on their level. However, it is important to recognize that for Stewart, Akamatsu, and Mayer, signing was just an imperfect vehicle for English acquisition. Mayer and Well (1996), as well as Mayer and Akamatsu (1999) had previously argued that natural signed languages were an imperfect and incomplete method of ensuring print language acquisition. This theme is repeated in later research, such as Mayer and Trezek (2020). The key point is that bad signing was acceptable, but bad English was not; it is where their arguments collapse under ableist thinking. Good communication requires a mutual respect of all available language and semiotic resources. If one language is prized over another, that is not communicating, it is indoctrination and neglects the reality of how people use language.

Much of this line of thinking about communication and languaging is rooted in beliefs that languages are discrete and bounded. However, as we know from *translanguaging* (Otheguy et al. 2015), fundamentally, languages are not discrete separate entities. Languages are a political category. Similarly, the United States exists as a political entity; the borders are formally drawn, and their violation is subject to various types of violence. However, on the borderlands, life is not so clear (Cantú 2011). Ideas such as Mexico or the United States are not so vividly defined; there are only the people, the environment, and the difficulties of navigating underneath the weight of these two distinct entities. The same is true of language and linguistic borderlands. In Cantu's borderlands, languages, such as English and Spanish, are not separate categories. Rather the languages overlap and merge until they are no longer separatable entities. This paper focuses on a kind of borderland that deaf people in the United States live in, though generalizations may be made to deaf people living in other countries. Deaf people are constantly in a state of language contact across signed languages, spoken languages, and print languages. The contacts are often strictly regulated and driven by ideology wherein hearing people try to enforce a very specific way of speaking and writing—one that is not *deaf*. Deaf people who are not forced to only speak often have exposure to many different signed systems and cue languages, many of which were developed because of the mistaken belief that natural signed language does not provide access to print (e.g., Crume et al. 2020). These translanguaging systems have been researched by various deaf scholars (e.g., Kusters 2021; Moriarty Harrelson 2017; Hou 2022). However, deaf people become adept translanguagers often in response to hearing people failing to craft language communities around them in the same way that they do for their hearing children. Translanguaging as a theoretical umbrella, however, is limited as it does not always address the interaction of language and disability. That is because translanguaging and applied linguistics scholars (e.g., Pennycook 2021) often did not see or recognize disability as a viable identity that can influence language in positive directions. For example, how does one translanguage when the language does not have access to a standard language in any modality? Deaf scholars, such as Erin Moriarty and Lina Hou (see multiple citations in this paper), have sought to answer this question. However, their scholarship does not percolate into densely hearing fields because hearing assumptions about the expanse of language can be limited.

## 2. Language Deprivation and Crip Linguistics

As a relatively recent theory, our goal in developing the Crip Linguistics theoretical framework was to provide an easily accessible umbrella for various theories of language through a critical disability framework. Crip linguistics unite different scholarly fields such as history, anthropology, disability studies, and literature among others. The core tenet of Crip Linguistics is that no one language is wrong; however, the stigma of disability is such that first, people look for ways to center white abled ways of languaging (e.g., speech), and second, people tend to assume that disabled people are automatically unable to language well without some kind of intervention and support. The consequence of structural ableism is language deprivation and the assumption that deaf children are languageless without linguistic toolkits.

Koulidobrova and Pichler (2021) used Crip Linguistics to take the idea that language-deprived deaf children have no linguistic toolkits to task, tackling the conundrum of the "Late L1 [first language] language learner." The Late L1 language learner is generally a deaf child who, for whatever reason, does not have a full language until they attend a signing school. Much of psycholinguist Rachel Mayberry's work in the 1990s and early 2000s focused on studying and defining the Late L1 language learner (e.g., Mayberry and Eichen 1991; Mayberry 1993, 2007; Mayberry et al. 2002; Mayberry and Lock 2003, among many, many more). The debate Koulidobrova and Pichler approached centered on how many linguistic resources Late L1 language learners bring to the table when they begin learning a standardized language. The problem Koulidobrova and Pichler (2021) needed to solve was if homesign systems transfer to a new L1 in similar ways that L1s transfer to L2s (second languages and on). Teachers of the deaf are often told when receiving a new student who has emigrated from countries without early intervention resources, that their students have no language. Sometimes their students actually know another standardized signed language (e.g., Lengua de Señas Mexicana; LSM) and whomever is providing information to the teachers assumes any signed language other than what they know is simply nonsense gestures. Often the students would be classified as *home signers*. The subtext here is that any homesign system does not function at all like a full language, at least not cognitively so, and that other signed languages are unrecognizable. However, by 2016, scholars, such as Coppola and Senghas (2016), were arguing that the distinction between a homesign system and a language were not so stark. Hou (2018) pointed out indirectly that *homesign* is misapplied in many situations by people who attend only to standardized and politically authorized language situations. Hou (2022) establishes that what is considered a language is deeply ideological. *Homesign*, for example, has been generalized across various non-standardized locally developed languages, although there does seem to be some allowances for the size of the community (e.g., village signed language; Hou and Kusters 2020).

Carrigan and Coppola (2017) showed that family members did not use family-based signed systems (often referred to as homesigns) enough to understand nuance. Parents simply did not learn well the gestural systems of their children. Hou's (2018) study highlights a key difference. San Juan Quiahijie Chatino families had multiple deaf people in each family. It seems then that increased numbers of deaf people in contact with each other and with hearing people incentivize the care work necessary to ensure quality communication with people. A standardized language does not matter in this case. The role of care work in languaging without a standardized language is apparent in Moriarty Harrelson's (2017) focus on the languaging of deaf people in Cambodia. Although considered to be without language, deaf Cambodians demonstrated an exceptional ability to use their semiotic repertoire to navigate the world and its communities. Language ideologies limit our understanding of what language is and does. Hou's theories are central to Koulidobrova and Pichler's (2021) argument. The Crip Linguistics framework validates deaf people who language using non-standardized languages. As they write, "A crip linguistics view may recognize impaired language, but impaired language should not be dismissed as "bad language", intrinsically disordered language, or, especially, non-language" (p. 10). This approach contradicts Mayberry's well documented work that non-standardized languages

(e.g., homesign) do not operate in the same way as L1s do. The answer to this conundrum exists in translanguaging theories. It is not clear how non-standardized languages influence the acquisition of standardized L1s because seeing this influence requires recognizing clear borders between the two languages, which simply do not exist under the translanguaging paradigm.

Deaf activists and educators who are supportive of deaf community movements would be quick to dismiss this discussion about the linguistic resources of those who do not use standardized signed languages. They feel concerned that the focus on validating various communication methods in the absence of a standardized language would minimize the need to build structures to support deaf children in developing language. The problem, they would likely argue, is *language deprivation*. Language deprivation and its effects have long been recognized in deaf communities (Robinson and Henner 2018). However, the term has become more salient with the work of psychologist Wyatte Hall. As he explains in his foundational paper in Hall et al. (2017), "language deprivation occurs due to a chronic lack of full access to a natural language during the critical period of acquisition (when there is an elevated neurological activity for language development), approximately in the first five years of a child's life" (p. 2). Over the past decade, work on countering language deprivation in deaf and hard of hearing children has become an industry and remains central to deaf community organizing in the Global North as it has since the nineteenth century (Murray 2015).

Deaf communities have recognized that language deprivation is typically paired with other types of abuse. Evidence for this can typically be found in discussions of the *Dinner Table Syndrome* (e.g., Hall et al. 2017; Meek 2020). The Dinner Table Syndrome basically describes how deaf people are shut out of typical family discourses as symbolized by the dinner table. At the dinner table, families talk about their day, argue about politics, and generally interact with each other. The deaf person, however, eats their dinner alone because they are not included. Sometimes a family member, usually the mother or a sibling, will make parts of the conversation accessible, but often the deaf person is told that the conversation will be summarized later. The people at the table even pretend that the deaf person heard what was being said and respond negatively if the deaf person cannot align with their expectations. The Dinner Table Syndrome is not about simple language deprivation, but being shut out of family dynamics, being ignored, and sometimes being subject to abuse because of a refusal to recognize the reality of a disability. When Jon was a trainee teacher of the deaf during his undergraduate study, he worked with one child who was struggling to learn the standardized language and English. The child's mother had several other younger children and worked full time. Her mother, the child's grandmother, cared for him along with his siblings. The grandmother did not know how to interact with the deaf child and placed him in front of the TV all afternoon and night until the mother picked up the children. While this alone is not a form of language deprivation, it was the complete lack of interaction, exposure to incidental language, and familial neglect that led to language deprivation. Given the complexities of reality, it is not surprising that the people who believe that the only valid language is oral, those who insist that deaf children learn to listen aurally and speak orally despite being deaf, have decided that the only solution to these issues is not to change the system so that families like these have the support they need to support the development of deaf children, but rather, make the deaf child as hearing as possible so that the capitalist machine churns unbothered (Robinson and Farah, forthcoming).

## 3. Policing Language in the Classroom: Who Can Judge Fluency?

In the Crip Linguistic framework, we recognize that non-normative bodies extend past racialization; accordingly, raciolinguistics alone is insufficient for analyzing language oppression. Language ideologies include policing how women speak (e.g., vocal fry and rising intonation), and also on gender-based language ideologies wherein the languaging of trans and non-binary people are discriminated against because they deviate from normative

expectations of how genders should language. In response, linguists such as Konnelly (2021) have proposed a *trans linguistics* that focuses on trans languaging in the same way that *Crip Linguistics* focuses on disabled languaging and *raciolinguistics* governs race in language attitudes. The link between trans linguistics and Crip Linguistics is stronger when one realizes that there is a growing body of research attempting to link trans identities to autism (Van Der Miesen et al. 2016). If there is a link, it is likely because autistic people, already marginalized because of their disability, do not feel the same social pressures to operate along strict gender ideologies. Nevertheless, the perceived link between gender identity and disability is enough for some people to see this as *a thing to fix* rather than a variation that needs to be supported.

In the deaf education classroom, much like in the inclusive classroom, these language ideologies are directed at students. In his foundational text, Rosa (2019) described how racialized students and staff in a high school were often marginalized, not just for their language, but for how their racialized status caused others to perceive their language. Brown students, for example, were often placed in English language learner programs even though they were born in the United States. Their proximity to Spanish was enough to cause schools concern. Incidentally, hearing children of deaf parents are often placed in speech therapies because their proximity to the broken or non-existent speech of their deaf parents makes professionals worry about their speech and spoken language development (see Zaborniak-Sobczak 2021 for a discussion on that). In both cases, the goal here is ensuring the development of a specific English grounded in standard abled white speech.

Teachers can and do delight in their management of their students' languaging. Syed (2022) reported that some teachers in Pakistan's higher education found fulfillment in correcting Urdu- and Sindhi-flavored English in their students' writing. Jon, for example, reports that his teachers would demand that any use of signed language in the classroom be accompanied by voices; non-voiced signing was verboten. However, both Syed (2022) and Cushing and Snell (2022) report that teachers are given strict instructions that they must be the language police for their students; otherwise, they fail both the students and the school. Syed notes that many other language teachers in their study often felt guilty for having to manage their students' languaging and having to enforce a monolingual English policy. Cushing and Snell explain that teachers are monitored and penalized for not creating environments where standardized English is encouraged. If non-standardized English is used in the classroom either by the students (without correction) or the teacher, the teacher is punished. As they write

> *This framing by the inspectorate shifts responsibility onto teachers for policing both the language of themselves and minoritized others, adding to institutional pressures concerned with linguistic performance which are historically embedded in England's schools. The 'ungrammaticality' of teachers' language was a widespread concern in reports. (p. 14)*

In other words, teachers are *obligated* to police language; otherwise, they are seen by the institutions as failing their students. In research on deaf children, the role of the teacher in policing English development is everywhere. Berent (2001), for example, writes that teachers are responsible for guiding the development of proper English in deaf students. Furthermore, much of the anti-bilingual/bimodal teaching movement is centered on whether good English can come from non-English sources (e.g., Mayer and Trezek 2020), which, of course, is wrapped up in ableist and racist notions of what good English is like. Deaf people are often aware that our presence in schools and programs for the deaf is limited because our bad English may be contagious to the students. Additionally, our inability to correct the speech of oralized deaf children to normative hearing standards is likely a contributing factor to our lack of presence in oral-focused programs.

Who and what is allowed to police the language of students is a fundamental question in many discussions of language policies and pedagogy. In spoken and written language assessment, the challenges of making an assessment that is not biased towards an aesthetic ideal language are well documented (e.g., Chalhoub-Deville 1995; Schaefer 2008). Sign

language assessments are new enough that justice discussion is still novel. However, signed language assessments, especially in the Global North, still center whiteness in their creation and evaluation. Part of the problem is the norming sample population of deaf signers is often not diverse.

This raises the question of who is a *native* speaker or signer. Those discourses also illustrate what a *native* speaker of English *should* look like. In psycholinguistics and linguistics research, a native speaker is often someone whose internal grammatical locus is tuned such that the data acquired from them in judgment tests are valid, which is a bit of a circular definition but seems to hold in linguistics research. Researchers are likely to create judgment tests that fit their own conventions of what languaging is or should be. In theory, nativeness is attributed automatically to people who have acquired language during critical periods (e.g., Piller 2002). In deaf-related research, nativeness is attributed to deaf children who have deaf parents (Conrad and Weiskrantz 1981; Orlansky and Bonvillian 1985). However, the idea of a native speaker becomes entwined with raciolinguistic ideologies. In the case of English, the idea of a native speaker is often restricted to those who are coded as white (Ramjattan 2019). As he writes, "Within the entire English language teaching (ELT) industry, white native speakers enjoy the privilege of being perceived as 'naturally qualified' teachers, due to the historical link between English and western imperialism, as well as the perception that adopting the speech of these individuals advances one's status in the globalized economy (Motha 2014; Phillipson 1992)." (p. 126). In this scope, whiteness and nativeness as English speakers are often inseparable. Not every white person is a native speaker of English, of course, but the issue here is that non-white people are more likely to be coded as non-native than white people based purely on raciolinguistic ideologies. Discussions about non-white native signers are often left out of the discourse because deaf education and linguistics researchers simply have not invested in accessing them. We are left to make assumptions on how deaf non-white children of deaf non-white parents are treated in the classroom based on how their hearing counterparts are treated.

Because teachers are obligatory police of classroom languages, they often fall back on raciolinguistic ideologies. Teacher training and licensing is designed to ensure that teachers who end up in the classroom are willing and able to police student languaging. This does require that the teachers themselves have the skills necessary to ensure that student languaging meets the needs of the state. In the United States, these skills are often measured by assessments such as the Praxis or performance-based assessments such as edTPA. Like many assessments, both the Praxis and edTPA are challenged by the variable performance of marginalized groups. A 2014 report by Ahmad and Boser (2014) for the Center for American Progress showed that 42.3% fewer black test takers passed the Praxis II English test compared to white test takers. Performance assessments such as the edTPA were supposed to address the injustice of assessments like the Praxis; however, an analysis from Petchauer et al. (2018) demonstrated that racial biases are still entrenched in the edTPA. Evidence of racial disparities in the edTPA existed across several years of analysis in the early 2010s. Overall, this means that licensed teachers tend to be whiter, with language ideologies that align with expectations that a certain type of English will be enforced in the classroom. Deaf education teacher prep students are also required to take assessments such as the Praxis and edTPA.

Older reports on the state of the deaf education field indicate that many people felt that Praxis and like assessments were necessary gatekeeping for the deaf education profession (e.g., Rosenfeld et al. 1994). However, even those reports were concerned that deaf or hard-of-hearing teachers would be gatekept via those tests because the kinds of Englishes used by deaf and hard-of-hearing people are not always in alignment with the preferred standardized Englishes. As Humphries (2013) and Baynton (1996) point out, deaf and hard-of-hearing people have historically been excluded from deaf education as teachers since the ascendancy of oralism in the late 19th century (p. 7). Simms et al. (2008) explained that the percentage of teachers of the deaf who were deaf and hard of hearing was entirely too small, and that the pool of teachers tended to be white.

No equivalent to Praxis or edTPA assessment for signed languages exists for teachers of the deaf in the United States. To the extent that assessments exist in other countries, they suffer from the same issues addressed herein. Some deaf education programs may require that their students take a national assessment such as the American Sign Language Proficiency Interview (ASLPI). Some of these details are enumerated in Humphries (2013), who described the ASL requirements of the deaf education program at the University of California, San Diego. However, these are program-level requirements and are not actually related to licensure. At the time of this writing, no national level data on teacher of the deaf signed language skills could be located. Some assumptions can be extrapolated from signed language interpreter licensing assessments. Earlier work from Schick et al. (1999, 2006) demonstrates that many signed language interpreters working in the classroom have passable conversational skills, but certainly not the skills required to mediate the classroom language environment. Given the racial disparities that exist on most assessments, one would assume that similar variations exist between the performance of marginalized license seekers and white ones. No easily retrieved data on exist at the time of this writing though. However, as detailed by Stewart (1993) and Akamatsu et al. (2002), signed language skills are simply not considered the equivalent of spoken and written language skills in both theory and application.

At this point, the reader may wonder what the theories described previously have to do with who can be a teacher of the deaf. The answer is everything. Language ideologies, racio and crip, permeate the classroom. We do not have a really good idea of what fluent signing looks like and are forced to use internalized models of what good signers are to assess signing. These models are often white, abled, and from language-rich environments that most deaf children do not have access to. Additionally, those assessing the signing have themselves not been assessed. Given that the desire is that teachers of the deaf sign well to deaf signers, the question of who can sign well, what it means to sign well, and why it is necessary to sign well persistently appear.

## 4. Who Can Teach the Deaf?

Fluency in signed languages is expected of teachers of the deaf to provide a scaffold for their students to attain proficiency in navigating a world otherwise hostile to non-speakers. The past two sections of this paper focused on building the background knowledge necessary to consider if teachers of the deaf needed to be fluent in signed languages. We have recognized that both teachers and students are under pressure to present a singular kind of language that is considered good. Additionally, what is good is subject to social pressure to be a specific kind of white abled spoken languager. Adhering to cripped linguistics theories means envisioning a radical new kind of languaging environment in the deaf education classroom, where translanguaging theories meets acceptance of non-normative ways of using language.

### 4.1. Position 1: Teachers of the Deaf Must Have Attained the Fluency in All Languages That Their Students Are Expected to Use to Pass Required Assessments

Given that language environments at home are highly variable and sometimes not completely accessible for deaf children, it is imperative that deaf children have an environment in which they can freely use and access language. Research does demonstrate that accessible language environments have positive benefits for deaf children (e.g., Henner et al. 2016). This does seem to contradict the point made previously in this paper. If there is no such thing as proper languaging, and fluency as a concept is wrapped up in raciolinguistic ideologies, then why would it be important for teachers of the deaf to language well and to language well using signed languages?

The answer to these questions lies in the difference between equity and justice, as it pertains to disability rights (see Sins Invalid 2015). First, the Crip Linguistics theory recognizes that language deprivation is a type of environmentally enacted impairment. It is something done to deaf people by hearing people. Secondly, Deaf people are punished

heavily on educational assessments. Although Traxler (2000) is over twenty years old at the time of this writing, the low assessment scores of deaf people for reading and math have remained relatively consistent today. In fact, the main argument from monomodalists (people who only support speaking and listening) such as Geers et al. (2017) and Tamati et al. (2022) is that signed languages contribute to poorer print literacy skills in deaf children. It is not an exaggeration to say that deaf education spins on the axis of print literacy; all other skills are minimized so that deaf children can access society using the preferred standardized language(s) of the government. Signing deaf children are obligated by the system to be proficient in the preferred standardized languages. When teachers of the deaf are only proficient in the preferred standardized languages, and not the languages that the deaf people use, then they have failed to provide an equitable and accessible environment for language learning (Humphries 2013). In sum, they have set their students up to fail; their students are required to be better at language than their teacher. That is not an equitable learning environment because hearing students are almost never in classrooms where they language better than their teachers.

However, all language requirements are intrinsically unjust. Good language in the United States is a proxy for proximity to whiteness. Good language requirements turn teachers and sometimes peers into language police. Racialized teachers will likely not be identified as having good language by the state; this applies to signed language assessment as well. Deaf teachers of the deaf are often unable to meet the licensure requirements of standardized English assessments. Currently, deaf teachers of the deaf are often given lee-way on standardized English requirements not given to other minoritized groups because there is research demonstrating that deaf children do respond well to deaf teachers even if their English may not be quite as good as their hearing co-workers (e.g., Andrews and Franklin 1997). This minor nod to justice is often done to the chagrin of hearing teachers who complain about a justice given to deaf people not given to them. As deaf scholars, we have so very rarely heard any recognition that the playing field for deaf people was already unlevel and perhaps support and advantages given to deaf people may not always be given equally to the hearing.

Yet, requiring teachers of the deaf to meet minimum language requirements does not enact crip justice, even though it is an equitable thing to do. Justice, however, demands that we either ignore or tear down the system. Justice allows us to envision a different way of educating deaf children. Towards this we argue the following:

*4.2. Position 2: Teachers of the Deaf Are Expected to Maintain a Level of Fluency That Provides a Scaffold for Their Students to Attain Proficiency at Navigating a World Otherwise Hostile to Non-Speakers*

If language requirements in the classroom are intrinsically unjust, what does a just teaching environment look like? Bell Hooks wrote the following about *transgressive classrooms*, " . . . a rethinking of ways of knowing, a deconstruction of old epistemologies, and the concomitant demand that there be a formation in our classrooms, in how we teach and what we teach, has been a necessary revolution—one that seeks to restore life to a corrupt and dying academy" (p. 27). The question for deaf education then becomes what does a deconstruction of old epistemologies look like? How do we adjust how we teach and what we teach, and how do we restore life to the corrupt and dying deaf education classroom? We have previously established that deaf children are prone to language deprivation wherein they are not exposed to the standardized languages of the community and often develop either extreme variations of the standardized language or they develop a non-standardized language. In either case, deaf children are then penalized for this in the school system. The best way to counter this according to researchers (e.g., Henner et al. 2016) is early entrance to an environment that provides standardized language in an accessible way. However, as discussed in the section on raciolinguistics, any kind of language requirement for becoming a teacher will exclude those who language in ways that are not considered normative by the system. In this case, it usually means those who use spoken language other than standard American English will be less likely to become teachers. In Rosa's

(2019) book, teachers who use community variations of English were called stupid. Yet, educational linguists such as Baker-Bell (2019), explain that in spite of "white eyes and ears" on community variations, students thrive when teachers speak their language. It seems like teachers need to first have mastered the expected standardized languages in order to pass the assessments so that they can end up in the classroom where they can use *the language of solidarity* (Baker-Bell 2019). What is the language of solidarity for deaf children without accessible language at home? The difference between Baker-Bell's classroom and the deaf ed classroom is that teachers of the deaf, while expected to master standard English, are not expected to master standard signed languages. However, here, we do not argue that good signing means that a person is able to satisfy signing assessments designed to measure fluency or proficiency in the standard dominant signed language. Those assessments are ideologically driven; they are more about the aesthetics of the ideal signer and how that ideal is rooted in whiteness and ableness. Additionally, the assessments do not measure the teacher's ability to navigate unbounded languages via a translanguaging framework.

In crip justice, what matters is the teacher's ability to facilitate language learning not just in English but in all possible language variations. The teacher must be a good model of calibration with an expansive semiotic repertoire, who understands that languaging requires active efforts to understand the other person. Being a fluent signer and excellent calibrator compensates for the absence of multimodal language models in the child's environments and communities outside the school. Part of this attitude includes embracing signed languages for their richness and potentiality rather than consigning them to the back burner in favor of English language acquisition. For example, deaf children have been punished for informing the teacher that the sign for butterfly was not a compound of the sign for dairy butter and then making a separate sign suggesting that the butter flies. In the crip classroom, those children would not be punished but instead rewarded for sharing language knowledge with the class and the teacher where everyone learned together. Instead, with punishment, students become resentful. The language learning of the classroom is sacrificed to preserve the feelings of the teacher. More importantly, students learn that their language has no value. In a just classroom, the language that the students use and their knowledge of it, regardless of any perceived fluency, is valuable. Teachers need to have the resources to learn, learn well, and create an environment that students and teachers can build off each other's language. This is the key point. We assume that once we are fluent in a language, we have reached its pinnacle. However, that is not how language works. All of us are students of a language for a lifetime. Teachers of the deaf must constantly learn how their students language and teach them how to build on crip competencies so that everyone in the classroom can language better.

## 5. Conclusions

At the beginning of this article, we discussed a television show directed at young children who had less than proficient signers interpreting spoken English songs for the benefit of the audience. While the judgment of their languaging was not rooted in crip justice, the reality is that deaf children need proficient linguistic models in order to access language capital (and other types of capital). Additionally, signed languages are the best vehicle for access to that capital for deaf children, including as a vehicle for the acquisition of the dominant majority written and spoken language, although we argue that written and spoken language proficiency should not be the sole goal of deaf education.

Unlike the signers on the children's TV show, deaf people would not be given the same leeway for the variation in their language ability in either English or signed languages, especially with such a public-facing job. Teachers of the deaf are expected to have a good command of the majority language in the written modality, and there is no parallel expectation of having a good command of signed languages. Teachers of the deaf who sign badly but write well obtain jobs; teachers who are fluent signers but write in average English (for example) are systemically barred from teaching through state assessments and licensure requirements that consider signed languages as invalid or, at least, not important. However,

in deaf education, crip justice is to prioritize deaf signers who can be expert models at calibration and pass on deaf community knowledge, as well as disability competencies. The system itself, at least in the United States, also has more testing requirements for deaf educators in the State-supported language, English, than the community-preferred language, ASL. It is accordingly socially acceptable to be an inept signer and a teacher of signing deaf children. The inverse is not true; it is not socially acceptable for a teacher of signing deaf children to be a proficient signer and an inept user of English. Here, we tread a fine line. How do we measure proficiency when assessments of signed languages are rooted in the aesthetic of a white deaf signer from a particular type of background (e.g., congenitally deaf people)? Production assessments of signed languages are usually locally crafted by a small group of people with a specific kind of signer in mind as the ideal model. This signer may not represent the diversity of signers of the community. Computerized assessments are rare, and while assessment may be more rigid compared to production assessments, these assessments are still built by people who have biases and these biases still favor a certain aesthetic of signers (e.g., Shohamy 2022).

If assessments can rarely be neutral, but assessment is necessary to promote language development in the classroom, then how do we ensure access to language and justice principles? There is always going to be friction between where a theory meets reality, and where equity meets justice. We hoped that by laying out our positions regarding teacher fluency, we could provide grounding for advancing assessment justice via the Crip Linguistics framework. Teachers should be fluent, but any assessment of fluency must be analyzed back and forth for biases. These biases must be confronted by the assessment developers and maintainers. Additionally, it has to be achieved with the understanding that any line drawn in the sand regarding who is a competent signer and who is not invariably discriminates against some marginalized identities. Assessments themselves also reify language borders, which can violate the natural translanguaging between modalities that deaf people have. The justice aspect is in how we respond to these situations. That will require more discussion in various deaf communities and may require that we give up many of the language ideals that we fought for, such as tight boundaries around natural signed languages.

**Author Contributions:** J.H. conceived the paper. J.H. and O.R. performed the analysis and wrote the paper together. All authors have read and agreed to the published version of the manuscript.

**Funding:** This research received no external funding.

**Institutional Review Board Statement:** Not applicable.

**Informed Consent Statement:** Not applicable.

**Data Availability Statement:** This publication follows COPE guidelines. No data is available as this is a theoretical paper.

**Conflicts of Interest:** The authors declare no conflict of interest.

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
