# Peer review of "Crip Linguistics Goes to School"

_languages, doi:10.3390/languages8010048_

Round 1
Reviewer 1 Report
This paper brings together several research frameworks in order to consider questions of equity and justice in deaf education. The author lays out how their theory, Crip Linguistics, pushes us to think about who can/should/will be in positions of more (and less) authority and power in the lives of deaf children. I take the main points to be:
1. Deaf children often encounter their first accessible language environment at school entry, meaning it’s especially important that this environment be linguistically (and otherwise) supportive.
2. Defining “good” language (or “fluent”) is inherently racist, classist, and ableist, which makes it challenging to think about what adults in the child’s environment need to be able to do to be linguistically supportive. Classifying people as having “good” language or “bad” language is not about objective linguistic practices but rather about identity.
3. Many bad faith arguments are found in this space that collapse once you require
“proficiency” in all languages/language varieties to be equally important.
4. Even with all of this said, there will always “be friction where a theory meets reality” (manuscript line 564).
This is incredibly important work and I love the low jargon descriptions of these intersecting frameworks, only some of which many readers will have prior familiarity with. This paper is aimed at practitioners, educators, and researchers and will make a great contribution for each.
I look forward to seeing this paper published but would like the author to provide a bit more detail in a few areas. I’d also love to see a more fleshed out conclusion. I believe these changes will be manageable over a relatively short timeframe and I recommend inviting this revision before publication.
General Points:
1. I would like to see a more extensive discussion of equity and justice in deaf education, especially at the end of this paper. The Conclusion section is quite short, so this might be a good place for this.
a. The author does define what they mean in terms of each earlier in the paper, but does so quite briefly. I think that subsections defining what the author means by each of these would be worthwhile.
b. I would love to see these points brought back together again at the end of the piece to give the reader more synthesis to mull over after reading. Definitely keep the point about how friction will remain here where justice meets equity, but I think this could fit nicely with the recommended more extensive discussion.
2. Can you say more about what impaired means within a Crip Linguistics framework? I understand what you are saying it does not mean (i.e. line 183-184, impaired isn’t a value judgment), but what does impaired mean in this framework? When do we use this term and what accompanying meanings do you want it to bring with it?
3. Relatedly, in Crip Linguistics what does it mean for “language development to happen well”? (Line 354-355). I would love to see this fleshed out.
4. I understand your point about oralism and capitalism (Line 239-243), but you don’t give the reader much background here. Can you refer back to another part of the paper or add detail? Or maybe combine with your points in Lines 254-257?
Small points:
(The irony of commenting on possible small writing tweaks many of which about “academic” writing style in a paper that’s all about how policing language isn’t something we should be doing is not lost on me. So ignore any of these you’d like to.)
1. Page 1, numbered list item 2: Number mismatch, either change to “social perception” or disorderïƒ disorders.
2. Very much appreciate this sentence as language contact is so often ignored in the case of signing: “Deaf people are constantly in a state of language contact.”
3. Line 135: united ïƒ unite
4. The use of “language” as a verb really works well, but I would recommend formally introducing that that’s what you’re doing since this will be novel for most readers. Maybe in a footnote on Page 3 where you first use language this way?
5. Line 149: much ïƒ many
6. Line 185: make this a bit stronger by substituting “contrast” with “contradict”
7. It seems to me that a lack of “proficiency” in the non-standard L1 (by the researchers) would also be a barrier to seeing influence on acquisition of the standard L1. Would be interesting to see you thoughts on this (Lines 188-190)
8. Tiny change: Very much agree with your point about Genie. I think she was locked in a bedroom though, not a bathroom (Line 211)
9. Line 225: Is a word/few words missing here? Should this read something like, “sometimes a family member, usually the mother…” or similar?
10. Line 233: Comma not needed
11. Line 270 has an extra word or a missing word: Either “some people perform at chance” or “some people do not perform above chance”
12. Lines 273-275: Love your points about masking and language development. You could also add that the cited studies are about college students, not young children, and these groups are of course quite different when it comes to language learning.
13. Line 367: Missing colon
14. Line 382: “accurate” here is about the ideas of those running oralist programs, is that right? Can you revise this so that readers are clear you’re not saying there’s an objective “accurate”?
15. Line 389: This definition is also circular in that a certain subset of people with a certain way of languaging create the judgement test that fits their (and only their) way of languaging, right?
16. Line 427: Excluded as teachers?
17. Line 447: Suggest changing both/and to either/or.
18. Line 462: Can you flesh out this transition a bit? Add another sentence or two about what the following section will aim to do?
19. Line 465: at home
20. Line 475: I’d repeat questions again “The answer to these questions lies…”
21. Lines 484-485: Excellent sentence (It is not an exaggeration to say that deaf education spins on the axis of print literacy)
22. Line 491: In sum
23. Line 496: requirements turn (not turns)
24. Line 543: Can you set up this anecdote more? Tell the reader what’s coming?
Author Response
Thank you for your thoughtful feedback, which was helpful in developing our paper. Our response to the reviewers are as follows.
Defining Impairment: We did not see the need to define impairment because our paper outlines disability as a social construct. An impairment is simply an injury, illness, or congenital condition that causes a difference of function, physically or psychologically. Disability is society’s interaction with a bodymind impairment. We follow and defer to critical disability studies discourses on usage of impairment, disability, and person first language/identity first language. However those conversations are complex and extensively covered across disability studies scholarship. We believe getting into this, for this paper, would be a distraction.
Defining the relationship between oralism and capitalism: this is an exceedingly complex conversation, for which we have a separate paper underway. The relationship between disability and capital is well covered in the scholarship. We have cited the forthcoming paper by Robinson & Farah.
We refrain from using footnotes as a point of access for blind readers. We have attempted to clarify some points within the text where possible as reviewers pointed out.
While we appreciated some of the reviewers’ respect for our paper in the spirit of crip linguistics, we strive for clarity when/where possible. We have addressed most of the points regarding clarity in our writing where you pointed out minor issues with conventions of grammar and punctuation. We have also added some transitions and clarified meanings of some concepts, e.g. oral(ist/ism).
We have fleshed out the following parts, as per reviewer comments:
- Conclusion
- Equity and justice as well as frictions in deaf education as well as questions about systems change.
We have added subsections fleshing out major terms: raciolinguistics, translanguaging, and language deprivation.
Thank you,
I look forward to seeing this paper published but would like the author to provide a bit more detail in a few areas. I’d also love to see a more fleshed out conclusion.
General Points:
- I would like to see a more extensive discussion of equity and justice in deaf education, especially at the end of this paper. The Conclusion section is quite short, so this might be a good place for this.
- The author does define what they mean in terms of each earlier in the paper, but does so quite briefly. I think that subsections defining what the author means by each of these would be worthwhile.
Conclusion re-written, subsections re-labeled.
- I would love to see these points brought back together again at the end of the piece to give the reader more synthesis to mull over after reading. Definitely keep the point about how friction will remain here where justice meets equity, but I think this could fit nicely with the recommended more extensive discussion.
- Relatedly, in Crip Linguistics what does it mean for “language development to happen well”? (Line 354-355). I would love to see this fleshed out.
Re-written to be made clearer.
- I understand your point about oralism and capitalism (Line 239-243), but you don’t give the reader much background here. Can you refer back to another part of the paper or add detail? Or maybe combine with your points in Lines 254-257?
We have attempted to make this clearer.
Small points:
(The irony of commenting on possible small writing tweaks many of which about “academic” writing style in a paper that’s all about how policing language isn’t something we should be doing is not lost on me. So ignore any of these you’d like to.)
Thank you for these minor corrections. Many were implemented, or were no longer needed as the sections pertaining to them were removed in our attempts to streamline the document.
Reviewer 2 Report
This was a very thought-provoking paper that could become a seminal text. I found myself grappling with the points raised and contextualizing them in the deaf education world we currently live in - a mark of some excellent scholarship. My only wish after reading the whole thing is to see section four fleshed out a bit more. One aspect I found myself wondering about in this section is with these points in consideration, what is to be done overall on a system level? What is to be done vis a vis hearing vs deaf teachers and other parts of deaf education that are so problematic? How should the system be dismantled, and what would the authors explicitly create in a post-crip world so to speak? I also encourage one more big-picture paragraph in the conclusion, which felt abruptly ended.
There were what I perceived as minor grammatical errors throughout but in the spirit of crip linguistics, I encourage the authors to make their own corrections as they feel fit rather than police it as a reviewer.
Author Response
Dear Reviewers,
Thank you for your thoughtful feedback, which was helpful in developing our paper. Our response to the reviewers are as follows.
Defining Impairment: We did not see the need to define impairment because our paper outlines disability as a social construct. An impairment is simply an injury, illness, or congenital condition that causes a difference of function, physically or psychologically. Disability is society’s interaction with a bodymind impairment. We follow and defer to critical disability studies discourses on usage of impairment, disability, and person first language/identity first language. However those conversations are complex and extensively covered across disability studies scholarship. We believe getting into this, for this paper, would be a distraction.
Defining the relationship between oralism and capitalism: this is an exceedingly complex conversation, for which we have a separate paper underway. The relationship between disability and capital is well covered in the scholarship. We have cited the forthcoming paper by Robinson & Farah.
We refrain from using footnotes as a point of access for blind readers. We have attempted to clarify some points within the text where possible as reviewers pointed out.
While we appreciated some of the reviewers’ respect for our paper in the spirit of crip linguistics, we strive for clarity when/where possible. We have addressed most of the points regarding clarity in our writing where you pointed out minor issues with conventions of grammar and punctuation. We have also added some transitions and clarified meanings of some concepts, e.g. oral(ist/ism).
We have fleshed out the following parts, as per reviewer comments:
- Conclusion
- Equity and justice as well as frictions in deaf education as well as questions about systems change.
We have added subsections fleshing out major terms: raciolinguistics, translanguaging, and language deprivation.
Thank you,
Authors
This was a very thought-provoking paper that could become a seminal text. I found myself grappling with the points raised and contextualizing them in the deaf education world we currently live in - a mark of some excellent scholarship. My only wish after reading the whole thing is to see section four fleshed out a bit more. One aspect I found myself wondering about in this section is with these points in consideration, what is to be done overall on a system level? What is to be done vis a vis hearing vs deaf teachers and other parts of deaf education that are so problematic? How should the system be dismantled, and what would the authors explicitly create in a post-crip world so to speak? I also encourage one more big-picture paragraph in the conclusion, which felt abruptly ended.
We have re-written the document to streamline things and to make our points clearer. Thank you for this feedback. To be honest we’re still working on many of the big picture things ourselves!
There were what I perceived as minor grammatical errors throughout but in the spirit of crip linguistics, I encourage the authors to make their own corrections as they feel fit rather than police it as a reviewer.
We truly appreciate this perspective.
Reviewer 3 Report
This is an interesting paper, although my impression at this stage was that it could be restructured and polished.
I strongly advise to make it clear and explicit if this text (or which parts of the text) is about the USA & ASL or about other SL communities or a globally applicable analysis.
English is not my first language, nevertheless I did find that the paper needs to be looked over for small spelling mistakes, punctuation, missing verbs and other mishaps, as well as style (word repetition, rather informal language, e.g. "the states" instead of US or USA) etc.
A few suggestions:
- Lines 68-70. There is an important statement about Spanish in the USA. Evidence for it is provided by reporting a fictional film/animated series. Why not quote (surely available) academic sources with empiric data instead of one anectodic example?
- Lines 77-78: A sentence for the transition between these paragraphs would be nice.
- Line 111: You might want to look into this wonderfully though provoking paper: Makoni, Sinfree, and Alastair Pennycook (2006) Disinventing and Reconstituting Languages. 1–41 in: Disinventing and Reconstituting Languages, ed. by Makoni & Pennycook, Clevedon: Multilingual Matters.
- Lines 166-173. I am lost. The text jumps too fast from one topic to the next.
- Lines 181-190. You are contrasting an entirely theoretical thought/approach with empirical work by Mayberry… I find that problematic and I actually don’t see the need to create a “contrast”. Both could be true.
- 198-200. I suggest quoting this differently. Omit the academic title and put all authors in the actual order so that your readers can locate the paper in the references.
- Line 209 ff. Do not explain to readers of this journal about Genie. Readers know.
- Line 240. Explain the term oralists. Many readers of this journal might not be familiar with it.
- Lines 242/243. While I understand what you are hinting, the leap from oralism to a few general words of capitalism critique without further explanation… seems reckless. Give more detail.
- Lines 263ff. I am not sure that the discussion of covid and masks is well placed in this paper. It seems to me that it distracts from the actual important points you are trying to make!
- Lines 278ff. Really no other source available than Twitter?!?
- Line 296. “racism and ableism is essentially the same thing” – that’s a big statement without any explanation or reference to an analysis.
- Line 314. I find the statement about “their natural language” troubling. Linking being Black to a “natural” language that comes with it is highly problematic. Maybe be more specific or quote the study in more detail.
- Line 354. I am certainly not a speech therapist, but I find that the statement is so overgeneralized (“never”) that it can only be wrong and will probably be perceived as rather hostile. Maybe change to “often”.
- Lines 399ff. Why not include a sentence on ELF. It’s a huge and important field of research and would fit here perfectly.
- Lines 426-428. The two quoted publications were published in the opposite chronological order (first Simms, Rusher, Andrews, and Corywell, then Humphries). It does not make sense to connect them by the word “additionally” when they are quoted in the wrong chronological order.
- Line 431 (“No equivalent assessment for signed languages exist for teachers of the deaf“). This is only true for the USA. Make that explicit.
- Chapter 4 starts with repeating the two main thesis. The chapter is structured with a chapter “4.1”, but I couldn’t find “4.2” anywhere. In general, section 4 is rather short and I would love to see more arguments and details here.
- It would be great to have more detailed and constructive conclusions and not new topics (“I discussed this a bit with Octavian Robinson”) introduced.
Author Response
Dear Reviewers,
Thank you for your thoughtful feedback, which was helpful in developing our paper. Our response to the reviewers are as follows.
Defining Impairment: We did not see the need to define impairment because our paper outlines disability as a social construct. An impairment is simply an injury, illness, or congenital condition that causes a difference of function, physically or psychologically. Disability is society’s interaction with a bodymind impairment. We follow and defer to critical disability studies discourses on usage of impairment, disability, and person first language/identity first language. However those conversations are complex and extensively covered across disability studies scholarship. We believe getting into this, for this paper, would be a distraction.
Defining the relationship between oralism and capitalism: this is an exceedingly complex conversation, for which we have a separate paper underway. The relationship between disability and capital is well covered in the scholarship. We have cited the forthcoming paper by Robinson & Farah.
We refrain from using footnotes as a point of access for blind readers. We have attempted to clarify some points within the text where possible as reviewers pointed out.
While we appreciated some of the reviewers’ respect for our paper in the spirit of crip linguistics, we strive for clarity when/where possible. We have addressed most of the points regarding clarity in our writing where you pointed out minor issues with conventions of grammar and punctuation. We have also added some transitions and clarified meanings of some concepts, e.g. oral(ist/ism).
We have fleshed out the following parts, as per reviewer comments:
- Conclusion
- Equity and justice as well as frictions in deaf education as well as questions about systems change.
We have added subsections fleshing out major terms: raciolinguistics, translanguaging, and language deprivation.
Thank you,
This is an interesting paper, although my impression at this stage was that it could be restructured and polished.
We have done this.
I strongly advise to make it clear and explicit if this text (or which parts of the text) is about the USA & ASL or about other SL communities or a globally applicable analysis.
We have done this.
English is not my first language, nevertheless I did find that the paper needs to be looked over for small spelling mistakes, punctuation, missing verbs and other mishaps, as well as style (word repetition, rather informal language, e.g. "the states" instead of US or USA) etc.
Anything written is intentional, including what is assumed to be informal language.
A few suggestions:
- Lines 68-70. There is an important statement about Spanish in the USA. Evidence for it is provided by reporting a fictional film/animated series. Why not quote (surely available) academic sources with empiric data instead of one anectodic example?
Pop culture is an important and empirical representation of social mores. We should not dismiss them as “unscientific”. Nevertheless we have removed this segment when streamlining the paper.
- Lines 77-78: A sentence for the transition between these paragraphs would be nice.
Ok.
- Line 111: You might want to look into this wonderfully though provoking paper: Makoni, Sinfree, and Alastair Pennycook (2006) Disinventing and Reconstituting Languages. 1–41 in: Disinventing and Reconstituting Languages, ed. by Makoni & Pennycook, Clevedon: Multilingual Matters.
Makoni and Pennycook did not discuss disability as an intersection of language analysis to the extent that is necessary. However, we did cite Pennycook’s recent adaptation of his applied linguistics book.
- Lines 166-173. I am lost. The text jumps too fast from one topic to the next.
- Lines 181-190. You are contrasting an entirely theoretical thought/approach with empirical work by Mayberry… I find that problematic and I actually don’t see the need to create a “contrast”. Both could be true.
We do not find this problematic. Empirical evidence is only reified via a theoretical framework. If the theory is deficient or could be improved, then the evidence may not be interpreted correctly. We don’t necessarily disagree with Mayberry. We think the results can be interpreted differently.
- 198-200. I suggest quoting this differently. Omit the academic title and put all authors in the actual order so that your readers can locate the paper in the references.
- Line 209 ff. Do not explain to readers of this journal about Genie. Readers know.
Readers may not know. Nevertheless we have removed this section.
- Line 240. Explain the term oralists. Many readers of this journal might not be familiar with it.
Done
- Lines 242/243. While I understand what you are hinting, the leap from oralism to a few general words of capitalism critique without further explanation… seems reckless. Give more detail.
Modified
- Lines 263ff. I am not sure that the discussion of covid and masks is well placed in this paper. It seems to me that it distracts from the actual important points you are trying to make!
Removed from our paper in streamlining
- Lines 278ff. Really no other source available than Twitter?!?
Twitter is a good source of community scholarship. We need to remove publication as a primary source of scientific research as marginalized people often are prevented from entering academia due to various systemic barriers (e.g. racism, ableism, etc). We cited TL Lewis’ twitter because we write from an activist disability stance: that is, disabled people confront many barriers in accessing academia and being published- especially if they are also black or brown. And so, we value community knowledge and contributions. Lewis is cited in Sami Schalk’s Black Disability Politics, which was released after we wrote this paper. We want to honor the original thinker and cite Lewis when possible. The relationship and logics of ableism and racism was clarified and a citation added for people who wish to better understand race and ableism.
- Line 296. “racism and ableism is essentially the same thing” – that’s a big statement without any explanation or reference to an analysis.
This comes from Talia Lewis’ work.
- Line 314. I find the statement about “their natural language” troubling. Linking being Black to a “natural” language that comes with it is highly problematic. Maybe be more specific or quote the study in more detail.
We have removed reference to the study. However, much of our language comes from scholars such as Dr. April Baker-Bell.
- Line 354. I am certainly not a speech therapist, but I find that the statement is so overgeneralized (“never”) that it can only be wrong and will probably be perceived as rather hostile. Maybe change to “often”.
If we are hostile, we intend to be.
- Lines 399ff. Why not include a sentence on ELF. It’s a huge and important field of research and would fit here perfectly.
We do not know what ELF is. The reviewer did not specify. We accordingly did not include.
- Lines 426-428. The two quoted publications were published in the opposite chronological order (first Simms, Rusher, Andrews, and Corywell, then Humphries). It does not make sense to connect them by the word “additionally” when they are quoted in the wrong chronological order.
We are not sure that this matters.
- Line 431 (“No equivalent assessment for signed languages exist for teachers of the deaf“). This is only true for the USA. Make that explicit.
Done.
- Chapter 4 starts with repeating the two main thesis. The chapter is structured with a chapter “4.1”, but I couldn’t find “4.2” anywhere. In general, section 4 is rather short and I would love to see more arguments and details here.
Revised
- It would be great to have more detailed and constructive conclusions and not new topics (“I discussed this a bit with Octavian Robinson”) introduced.
Noted.
Round 2
Reviewer 2 Report
I have no further edits. This is well done.
Reviewer 3 Report
yes, the mansucript has been drastically improved and can be published.